# Increased Invasion Risk of *Tagetes minuta* L. in China under Climate Change: A Study of the Potential Geographical Distributions

**DOI:** 10.3390/plants11233248

**Published:** 2022-11-26

**Authors:** Yuhan Qi, Xiaoqing Xian, Haoxiang Zhao, Rui Wang, Hongkun Huang, Yanping Zhang, Ming Yang, Wanxue Liu

**Affiliations:** 1State Key Laboratory for Biology of Plant Diseases and Insect Pests, Institute of Plant Protection, Chinese Academy of Agricultural Science, Beijing 100193, China; 2Rural Energy and Environment Agency, Ministry of Agriculture and Rural Affairs, Beijing 100193, China

**Keywords:** *Tagetes minuta* L., climate change, potential geographical distributions, MaxEnt model, invasive alien species

## Abstract

*Tagetes minuta* L., a member of the *Tageftes* genus belonging to the Asteraceae family, is a well-documented exotic plant native to South America that has become established in China. In this study, 784 occurrence records and 12 environmental variables were used to predict the potential geographical distributions (PGDs) of *T. minuta* under current and future climatic changes using an optimized MaxEnt model. The results showed that (1) three out of the twelve variables contributed the most to the model performance: isothermality (bio3), precipitation in the driest quarter (bio17), and precipitation in the warmest quarter (bio18); (2) the PGDs of *T. minuta* under the current climate covered 62.06 × 10^4^ km^2^, mainly in North, South, and Southwest China; and (3) climate changes will facilitate the expansion of the PGDs of *T. minuta* under three shared socioeconomic pathways (SSP 1-2.6, SSP2-4.5, and SSP5-8.5) in both the 2030s and 2050s. The centroid of suitable habitats under SSP2-4.5 moved the longest distance. *T. minuta* has the capacity to expand in China, especially in Yunnan, where there exist no occurrence records. Customs, ports, and adjacent regions should strengthen the quarantine of imported goods and mobile personnel for *T. minuta*, and introduced seedlings should be isolated to minimize their introduction risk.

## 1. Introduction

At the 15th Conference of the Parties to the Convention on Biological Diversity (CBD COP15) in 2021, biological invasions were discussed as one of the most challenging global issues of the 21st century [1]. Invasive alien species (IAS) can alter distribution patterns [2,3,4], thereby reducing the diversity of native species through interspecific competition [5,6]. Invasive alien plants (IAPs) are a critical part of IAS. With rapid global changes, the expansion of IAPs has not yet shown any indications of saturation. The number of IAPs will increase by an average of 18% between 2005 and the next 50 years [7]. Climate warming is the most vital factor in determining the extent of IAP colonization [8,9]. Climate warming will facilitate the spread and establishment of IAPs and change their hierarchies in ecosystems [10]. Amongst IAPs, Asteraceae is globally invasive owing to its biological characteristics, such as asexual reproduction and short growth periods [11,12], and one of the most problematic weeds in Asia, Oceania, and Africa, with serious ecological impacts and economic losses [13]. For instance, the invasion of Asteraceae weeds in America has reduced crop and pasture yields on agricultural lands [14]. To date, 58 species of Asteraceae have been identified in China, 12 of which have been included in the list of IAS in natural ecosystems in China [15,16]. Asteraceae seeds have invaded China mainly via wind, rivers, animals, and transport, causing over two billion ecological risks and industrial losses [17]. 

*Tagetes minuta* L. (Asteraceae: *Tagetes*), a widespread weed worldwide, is native to South America (Brazil, Argentina, and Peru), spreading to North America, South Europe, South Asia, Africa, Madagascar, and Australia [18,19]. Wild populations of *T. minuta* were first reported in Beijing, China, in 2011 [20]. Currently, it is generally widely distributed in North China and Tibet [21,22]. Well documented as a noxious plant, *T. minuta* was intercepted by plant quarantine customs in China in 2021 (GACC, http://www.customs.gov.cn/, accessed on 28 September 2022). The presence of *T. minuta* in China has the potential to cause the community homogenization of native plants, economic losses due to crop losses, and skin irritation in humans and fauna [23,24]. For instance, it has been shown that *T. minuta* is more competitive than Tibetan barley and thus invades it, affecting its normal growth and significantly increasing its mortality [25]. To date, studies on *T. minuta* in China have mainly focused on its biological characteristics [26], genetic evolution [27], and hazard investigations [6]. Currently, the potential geographic distributions (PGDs) of *T. minuta* are unknown; thus, the study of PGDs of *T. minuta* can provide an early warning of its further spread and has become an urgent research issue. 

Species distribution models (SDMs) are used to study species distributions that integrate specific species with ecological niche factors; they are currently widely used in autecology and biogeography [28,29]. The MaxEnt (Maximum Entropy Model) has become the most applied species distribution model owing to its advantage of using only existing occurrence records and its high performance with small samples [30,31,32,33]; moreover, it has gained popularity in studies on the PGDs of IAPs in recent years. For instance, MaxEnt models were used to study the PGDs of the IAPs *H. suaveolens* in India [34] and *S. alterniflora* in China [35]. Exceptionally, avoiding the overfitting disadvantages of the MaxEnt model built on default parameters and optimizing the parameters in R software [36] can effectively improve the reasonableness and accuracy of PGD predictions.

In this study, we aimed to predict the PGDs of *T. minuta* in China under current and future climate scenarios using the optimal MaxEnt model based on the global occurrence records of *T. minuta* and related environmental variables. We propose the hypothesis that the most suitable areas for *T. minuta* to colonize in China may not only be those where occurrence records are extant. Thus, we (1) determined the significant environmental variables affecting the PGDs of *T. minuta*; (2) modeled the PGDs of *T. minuta* in China under current and future climatic conditions; and (3) analyzed the changes in the PGDs of *T. minuta* in China under climate change. The solution to these issues will help understand how *T. minuta* successfully colonizes and expands rapidly in new habitats, further providing a targeted scientific basis for the prevention of *T. minuta* in China.

## 2. Results

### 2.1. Optimization Model

Based on 784 occurrence records of *T. minuta* and 13 environmental variables, the MaxEnt model was optimized using the ENMeval package to obtain the best prediction of *T. minuta* PGDs. The accuracy of the model was examined using the area under the receiver operating characteristic (ROC) curve (AUC). The results showed that the feature combination (FC) set to LQPTH and the regularization multiplier (RM) set to 0.5 were the optimal parameters in this simulation. The mean AUC was 0.956 for the 10 replications with this parameter, and the mean AUC values were 0.953, 0.954, 0.953, 0.953, 0.953, and 0.953 for projected climate change in the future. The simulation accuracy of predicting the PGDs of *T. minuta* in China using the optimized MaxEnt model was exceptional (Figure 1).

### 2.2. Significant Environmental Variables

Overall, the key environmental variables influencing the PGDs of *T. minuta* were the temperature elements (Bio2, Bio3, and Bio8), precipitation elements (Bio12, Bio15, Bio17, Bio18, and Bio19), topographic elements (Altitude), and soil elements (T_Clay, T_Gravel, T_Sand, and T_OC). During model fitting, the contribution represented the importance of environmental variables to the PGDs of *T. minuta* (Figure 2a). The top three variables with the highest contribution percentage were isothermality (Bio3, 34.8%), precipitation in the driest quarter (Bio17, 23.5%), and precipitation in the warmest quarter (Bio18, 11.4%), with a cumulative contribution of 69.7%. The results of Jackknife showed that the three variables that had the most significant influence on regularization training gain when using only the individual variables of isothermality (Bio3), precipitation in the driest quarter (Bio17), and precipitation in the coldest quarter (Bio19), indicating that these variables were more significant than the others (Figure 2b).

The relationship between the existence probability of *T. minuta* and environmental variables can be further clarified by the response curves (Figure 3), which were generally considered to be favorable for the growth of *T. minuta* when the probability of the existence of a habitat was greater than 0.5 (i.e., a high-suitability habitat). According to the response curves of the environmental variables, the isothermal range suitable for the growth of *T. minuta* was 45–51, precipitation in the driest quarter was 90–277 mm, precipitation in the warmest quarter was 177–2500 mm, and precipitation in the coldest quarter was 92–291 mm and 348–384 mm.

### 2.3. Occurrence Records and PGDs in China under the Current Climate

Native to South America, *T. minuta* was first discovered in 2011 in Beijing, China, and it has been successfully established in Beijing; Hebei, Shanxi, Jiangsu, Shandong, and Xizang provinces; and Taiwan, with a total of 30 recorded occurrences (Figure 4a). Figure 4b shows the PGDs of *T. minuta* under the current climate. In general, the majority of the PGDs of *T. minuta* under the current climate were mainly in the Beijing–Tianjin–Hebei region, Shanxi, and most of Yunnan Province, with smaller portions in Shandong, Inner Mongolia, Xizang, Guizhou, Sichuan, Fujian, Guangdong Province, and Taiwan. Specifically, the high-suitability habitat area was 9.39 × 10^4^ km^2^, accounting for 0.92% of China, mainly in northwestern Beijing, central Hebei, eastern Shanxi, and northern Yunnan provinces. The moderate-suitability habitat area was 20.52 × 10^4^ km^2^, accounting for 2.19% of China, mainly in eastern Beijing, southwestern Hebei, central Shanxi, and most of Yunnan Province. The low-suitability habitat area was 32.15 × 10^4^ km^2^, accounting for 3.35% of China, mainly in Beijing, Hebei, and Shanxi provinces, and central Shandong, Xizang, southern Fujian, Guizhou, northern Guangdong, northeastern Inner Mongolia, and Taiwan. Unsuitable habitats were found to be widely distributed in many provinces of China.

### 2.4. PGDs and Changes under Climate Change

Three shared socioeconomic pathways (SSP1-2.6, SSP2-4.5, and SSP5-8.5) were used to assess the impact of the projected climate change on the PGDs of *T. minuta* in the 2030s and 2050s. The results showed that climate change increased the distribution range of *T. minuta* in all three pathways. Overall, changes from unsuitable habitats to low-suitability habitats and from moderate-suitability habitats to high-suitability habitats were more pronounced than other changes in the projected climate in the 2030s and 2050s under SSP1-2.6, SSP2-4.5, and SSP5-8.5. Low-suitability habitats expanded mainly in Shandong, Shaanxi, Qinghai, Sichuan, Guangdong, and Fujian provinces, while highly suitable habitats expanded mainly around the Beijing–Tianjin–Hebei region and Yunnan Province (Figure 5).

During the 2030s, under SSP1-2.6, the highly suitable habitat area of *T. minuta* was predicted to be 13.12 × 10^4^ km^2^, the moderate-suitability habitat area was predicted to be 21.79 × 10^4^ km^2^, and the total habitat area was predicted to be 80.61 × 10^4^ km^2^, accounting for 1.37%, 2.27%, and 4.76% of China, respectively. During the 2050s, under SSP1-2.6, the high-suitability habitat area of *T. minuta* was predicted to be 17.99 × 10^4^ km^2^, the moderate-suitability habitat area was predicted to be 15.61 × 10^4^ km^2^, and the total habitat area was predicted to be 83.11 × 10^4^ km^2^, accounting for 1.87%, 1.63%, and 5.16% of China, respectively (Figure 6). From the 2030s to the 2050s, under SSP1-2.6, low-suitability habitats in North China and Southwest China and highly suitable habitats in the northwest of Yunnan Province showed a growing trend. From the present to the 2030s, the area where unsuitable habitats shifted to low-suitability habitats was predicted to be 22.58 × 10^4^ km^2^, and the area where moderate-suitability habitats shifted to high-suitability habitats was predicted to be 5.38 × 10^4^ km^2^. From the 2030s to the 2050s, the area in which unsuitable habitats shifted to low-suitability habitats was predicted to be 20.83 × 10^4^ km^2^, and the area where moderate-suitability habitats shifted to high-suitability habitats was predicted to be 7.09 × 10^4^ km^2^ (Figure 7).

During the 2030s, under SSP2-4.5, the high-suitability habitat area of *T. minuta* was predicted to be 2.27 × 10^4^ km^2^, the moderate-suitability habitat area was predicted to be 20.15 × 10^4^ km^2^, and the total habitat area was predicted to be 87.09 × 10^4^ km^2^, accounting for 2.27%, 2.1%, and 6.74% of China, respectively. During the 2050s, under SSP2-4.5, the high-suitability habitat area of *T. minuta* was predicted to be 18.62 × 10^4^ km^2^, the moderate-suitability habitat area was predicted to be 17.09 × 10^4^ km^2^, and the total habitat area was predicted to be 68.83 × 10^4^ km^2^, accounting for 1.94%, 1.78%, and 3.45% of China, respectively (Figure 8). From the present to the 2030s to the 2050s, under SSP2-4.5, high-suitability habitats showed a marked increase, whereas low-suitability habitats showed an increase followed by a decrease. From the present to the 2030s, the area where moderate-suitability habitats shifted to high-suitability habitats was predicted to be 10.48 × 10^4^ km^2^, and from the 2030s to the 2050s, the area where moderate-suitability habitats shifted to high-suitability habitats was predicted to be 4.22 × 10^4^ km^2^ (Figure 7).

During the 2030s, under SSP5-8.5, the high-suitability habitat area of *T. minuta* was predicted to be 16.88 × 10^4^ km^2^, the moderate-suitability habitat area was predicted to be 20.44 × 10^4^ km^2^, and the total habitat area was predicted to be 89.54 × 10^4^ km^2^, accounting for 1.76%, 2.13%, and 5.44% of China, respectively. During the 2050s, under SSP5-8.5, the high-suitability habitat area of *T. minuta* was predicted to be 15.42 × 10^4^ km^2^, the moderate-suitability habitat area was predicted to be 20.54 × 10^4^ km^2^, and the total habitat area was predicted to be 71.76 × 10^4^ km^2^, accounting for 1.6%, 2.14%, and 3.73% of China, respectively (Figure 6). From the present to the 2030s to the 2050s, under SSP5-8.5, high-suitability habitats in Yunnan Province showed a significant increase, while the others showed no significant differences. Moreover, the trends in changes in suitable habitats were roughly compatible with those under SSP2-4.5. From the present to the 2030s, the area where moderate-suitability habitats shifted to high-suitability habitats was predicted to be 7.97 × 10^4^ km^2^, and from the 2030s to the 2050s, the area where moderate-suitability habitats shifted to high-suitability habitats was predicted to be 4.18 × 10^4^ km^2^ (Figure 7).

In summary, under the three shared socioeconomic pathways, the PGDs of *T. minuta* expanded compared to PGDs under the current climate, mainly in low-suitability habitats in North China and South China and in high-suitability habitats in Yunnan Province. In particular, the most significant expansion occurred under SSP2-4.5. The suitable habitat area of *T. minuta* was significantly larger under SSP2-4.5 than under SSP1-2.6 or SSP5-8.5. In addition, the smallest increase was observed under SSP1-2.6.

### 2.5. Centroid Distributional Shifts under Climate Change

The centroids of suitable habitats of *T. minuta* are shown in Figure 9. Under the current climate, the centroid of suitable habitats was located at the point (107.79° E, 30.9° N). Under SSP1-2.6, the centroid of the suitable habitats shifted from the present to the point (107.51° E, 31.71° N) in the 2030s and then to the point (108.38° E, 30.83° N) in the 2050s; it shifted 0.28° E and 0.81° N from the current state to the 2030s and 0.87° E and 0.88° N from the 2030s to the 2050s. Under SSP2-4.5, from the current state to the 2030s, the centroid of suitable habitats shifted to the point (108.24° E, 31.41° N) and then to the point (106.35° E, 30.51° N) in the 2050s. It shifted 0.45° E and 0.51° N from the current state to the 2030s and shifted 1.89° E and 0.9° N from the 2030s to the 2050s. Under SSP5-8.5, from the current state to the 2030s, the centroid of the suitable habitats shifted to the point (107.87° E, 31.19° N) and then shifted to the point (107.19° E, 30.74° N) in the 2050s; it shifted 0.1° E and 0.29° N from the current state to the 2030s and shifted 0.68° E and 0.45° N from the 2030s to the 2050s (Figure 8). 

In summary, under the three shared socioeconomic pathways, the centroid of suitable habitats of *T. minuta* showed a general trend of shifting northward in the 2030s and southward in the 2050s. The centroid of suitable habitats under SSP2-4.5 moved the longest distance, the centroid of high-suitability habitats under SSP1-2.6 moved the second longest distance, and the centroid of low-suitability habitats under SSP5-8.5 moved the shortest distance.

## 3. Discussion

Changes in the climate, soil, and topography alter the distribution patterns of IAS [37], and as global warming continues, predicting the PGDs of IAS could facilitate the effective interception of global IAS invasion into native regions [7]. *Tagetes minuta* L. is considered a damaging IAS owing to its high sensitization and the decrease in native biodiversity after its colonization [25]. In addition, it has a strong reproductive capacity and high tolerance to the environment [12]. For instance, a study in Nyingchi, Tibet, showed that *T. minuta* belongs to a generalized pollination system and has successfully used local pollinators for pollination [25]. To prevent the introduction and colonization of *T. minuta*, which poses a threat to biodiversity, agricultural production, and human health, early monitoring and warning should be performed. This study is the first, and our results provide a scientific basis for the early monitoring and invasion of *T. minuta* in China.

### 3.1. Environmental Variables Influenced PGDs of T. minuta

The PGDs of *T. minuta* were subjected to a combination of variables, including temperature, precipitation, soil, and altitude. Our results demonstrate that *T. minuta* is often distributed in places with small annual temperature fluctuations and a warm and humid climate. From the perspective of temperature, our results showed that isothermality (Bio3) contributed the most and even played a major role in the model. Isothermality (Bio3) represents the steady state of temperature changes over a year. The response curves under the influence of univariate variables indicated that the suitability index of *T. minuta* was higher in regions where isothermality was in the range of 45–51; that is, when isothermality was less than 20, the survival probability of *T. minuta* dropped sharply, almost to zero, indicating that *T. minuta* cannot tolerate high-temperature changes, which is consistent with previous studies [38]. In addition, high-temperature stress, low-temperature stress, and severe temperature change stress will cause the degradation of chlorophyll in the leaves of *T. minuta*, and the enzymatic activities of soluble sugars and peroxides in the leaves will decrease [39]. From the perspective of precipitation, precipitation in the driest quarter (Bio17) and precipitation in the warmest quarter (Bio18) were also extremely important. If the precipitation in one quarter was less than 90 mm, the survival probability of *T. minuta* tended to zero, indicating that *T. minuta* was intolerant to drought and preferred humid or wet environments, which is consistent with previous research on the effects of high-precipitation conditions on the biological activity of *T. minuta* [40]. In addition, water affects plant growth, leaf traits, and the photosynthetic rate; if water is insufficient, root and stem growth would be inhibited, root biomass would be reduced, and it would also lead to the disruption of plant photosynthetic metabolism [41]. From the perspective of soil and altitude, our results showed that the contribution rate of most soil variables and altitude ranked after bioclimatic variables, indicating that *T. minuta* has extremely high adaptability to different soils (sandy, loamy, and clay) and reasonable altitudes in the tropics and subtropics [42].

In summary, *T. minuta* preferred slight temperature fluctuations, could tolerate precipitation up to 2500 mm, and showed high soil and altitude adaptability. Temperature and precipitation had a significant influence on the survival of *T. minuta*, whereas the soil environment and altitude played a secondary role.

### 3.2. Changes in T. minuta PGDs

Based on the occurrence records of *T. minuta* and the optimized MaxEnt model, our results showed that the PGDs of *T. minuta* in China under the current climate were mainly located in Beijing, Hebei, Shandong, Shanxi, and Yunnan, where no record of occurrence is currently available. According to the division of the global climate zone by the Köppen climate classification [43], the climate type in Yunnan Province is a humid subtropical monsoon climate, which is similar to that in the lowlands of northern India, Nepal in Asia, northern Argentina in South America, and South Africa, Angola, and Zambia in Africa, which are typical regions where the occurrence of *T. minuta* has been recorded [44,45,46]. Therefore, a valuable discovery was that Yunnan Province has become a high-suitability habitat for the invasion of *T. minuta* in China and a high-risk region that deserves vigilance.

Under future climate scenarios, the area of PGDs generally showed an increasing trend. There has been a similar conclusion in China that climate warming promotes an increase in the suitable area for *Asteraceae* plants, and it has been proven that the suitable area for *Bidens frondosa* L. will increase under the three scenarios in the 2050s [47]. However, the difference between the previous results and those of this study is that the spread area of *B. frondosa* under SSP5-8.5 was higher than that under SSP2-4.5. This may be because *B. frondosa* is a moisture-loving and drought-fearing plant that is more sensitive to precipitation, which is positively correlated with greenhouse gas emissions [48]. Until the 2050s, the spread of *T. minuta* in China mainly manifested in two directions. One is the expansion of low-suitability habitats to higher latitudes, such as Shaanxi, Gansu, Qinghai, and parts of eastern Inner Mongolia, which transformed areas of low-suitability habitats. Second, the area of transition from moderately suitable habitats to high-suitability habitats in the Beijing–Tianjin–Hebei region, as well as Shanxi Province and Yunnan Province, increased significantly. This validates the biology of *T. minuta* as being susceptible to reproductive expansion [27] and confirms the necessity of our research. The spread directions of *T. minuta* under the shared socioeconomic pathways of SSP1-2.6, SSP2-4.5, and SSP5-8.5 were roughly the same, but the spread area in the northwest direction under SSP5-8.5 was significantly smaller than that under SSP2-4.5, which may be because the increase in temperature and precipitation under the SSP5-8.5 condition exceeds the suitable growth range of *T. minuta*, and it is more likely to exacerbate the phenomenon of habitat fragmentation of *T. minuta*, which will have negative effects on its normal growth [40].

### 3.3. Strategies for Early Warning of T. minuta Invasion

From the perspective of introduction, the global spread pathways of *T. minuta* are mainly natural, global trade, and accidental [49,50]. Of these, accidental spread is the most common through pollinators, plants, crops, containers, wood products, soils, and human-related waste (CABI 2022). For instance, previous studies have shown that invasive *T. minuta* can use native pollinators, including bees, flies, and gophers [25]. In view of this, to prevent the introduction of *T. minuta* in PGDs, Beijing, Shijiazhuang, Jinan, Taiyuan, Fuzhou, Guangzhou, and Kunming customs should strengthen the quarantine of imported containers and wood products, especially for North America, South Europe, South Asia, Africa, and Australia. In addition, if *T. minuta* is introduced through customs located in non-PGDs, it should be exterminated in time to prevent its transfer to PGDs. From the perspective of colonization, Beijing, Shijiazhuang, Jinan, Taiyuan, Fuzhou, Guangzhou, and Kunming customs should be better managed, especially in Kunming, where no *T. minuta* invasion has been recorded. From the perspective of prevention and control, if a wild population of *T. minuta* is found in China, cultural, mechanical, and chemical control measures should be taken immediately to eradicate it (CABI 2022). Cultural control measures mainly involve uprooting, removal by hand, or mechanical cultivation [51]. Mechanical control measures, including tillage and hand pulling, are highly effective in controlling *T. minuta* in agricultural fields and cultivation processes [52]. Chemical control measures are more widely used to prevent *T. minuta* from accessing an area [53], such as acifluorfen, cyanazine, 2,4-D, and simazine. Finally, a comprehensive warning strategy for the early introduction, colonization, and control of *T. minuta* invasion was established.

## 4. Materials and Methods

### 4.1. Occurrence Records of T. minuta

Global occurrence records of *T. minuta* were collected from the Global Biodiversity Information Facility (GBIF, https://www.gbif.org/, accessed on 13 July 2022), Invasive Species Compendium of the Center for Agriculture and Bioscience International (CABI-ISC, https://www.cabi.org/isc, accessed on 13 July 2022), Atlas of Living Australia (ALA, https://www.ala.org.au/, accessed on 13 July 2022), Chinese Virtual Herbarium (CVH, http://www.cvh.ac.cn/, accessed on 13 July 2022), and our field survey. Finally, we obtained 1156 occurrence records of *T. minuta* from these online databases. Duplicate occurrence records and points without detailed geolocations were removed from the dataset. ENMTools (http://purl.oclc.org/enmtools, accessed on 15 July 2022) was used to screen the occurrence records of *T. minuta* for model simulation. To maintain consistency with the resolution of the environmental variables, only one occurrence record was retained within each 5 km × 5 km raster. Finally, 784 valid occurrence records for *T. minuta* were retained (Figure 9).

### 4.2. Environmental Variables 

Nineteen current bioclimatic variables (1970–2000) and altitude variables were downloaded from the World Climate Database (http://www.worldclim.org//, accessed on 13 July 2022), with a resolution of 2.5′ (Appendix A). Twelve soil variables were downloaded from the Harmonized World Soil Database (https://iiasa.ac.at/models-and-data/harmonized-world-soil-database; accessed on 13 July 2022). Future climate data were obtained using the BCC-CSM2-MR global climate model developed by the National Climate Center for two periods (the 2030s and 2050s) and three shared socioeconomic pathways (SSP1-2.6, SSP2-4.5, and SSP5-8.5) (Figure 10). The world administrative map was downloaded from the National Earth System Science Data Center, National Science and Technology Infrastructure of China (http://www.geodata.cn, accessed on 13 July 2022), and MaxEnt 3.4.4, which is freely available online (http://biodiversityinformatics.amnh.org/open_source/MaxEnt/, accessed on 13 July 2022).

There may be a linear correlation between the 35 environmental variables associated with *T. minuta* occurrence records. A correlation test (Pearson’s) of 35 environmental variables was performed using ENMTools (Appendix A). The process of shortlisting consisted of two steps: (1) 35 environmental variables were imported into the MaxEnt model three times, and those with zero contribution were removed; (2) the residual environmental variables with a contribution of more than zero were subjected to correlation analysis in ENMTools, and when the correlation coefficient between two environmental variables was more than or equal to 0.8, the variable with the highest contribution was retained. Ultimately, 13 environmental variables were retained (Bio2, Bio3, Bio8, Bio12, Bio15, Bio17, Bio18, Bio19, Altitude, T_Gravel, T_Sand, T_Clay, and T_OC).

### 4.3. Model Settings and Evaluation

As the most important parameters of the MaxEnt model, the calibration of FCs and the RM can significantly improve the prediction accuracy of the model [54,55]. There were 48 different combinations of the five basic parameters: linear-L, quadratic-Q, product-P, threshold-T, and hinge-H. The RM was set to 4 or less and used an interval of 0.5, increasing from 0.5 to 4, for a total of eight values in this study. The ENMeval package in R software (https://www.r-project.org/, accessed on 13 July 2022) was used to create 48 candidate models [56]. Finally, models with significant delta values were selected. AICc values were equal to 0.

### 4.4. Suitable Habitat Classification 

The generated ASCII raster format was converted into raster format in ArcGIS software (https://www.arcgis.com, accessed on 13 July 2022) and extracted according to the administrative division map of China. Based on the maximum test sensitivity and specificity cloglog threshold, habitats were classified into four potential categories: high-suitability, moderate-suitability, low-suitability, and unsuitable habitats.

### 4.5. Centroid of PGDs

Using Statistical Analysis Zonal in ArcGIS software, we could scientifically and intuitively capture the direction and distance of the changes in the PGDs of IAS [57]. The formulas are as follows:*Xt* = *i* = 1m(*Cti* × *Xi*)/*i* = 1m*Cti*(1)
*Yt* = *i* = 1m(*Cti* × *Yi*)/*i* = 1m*Cti*(2)
where *Xt* and *Yt* indicate the latitude and longitude, respectively, of PGDs in period *t*; *Cti* indicates the area of the i-th PGD in period *t*; *Xi* and *Yi* indicate the latitude and longitude, respectively, of the *i*-th PGD plaque; and m is the total number of plaques of the *i*-th PGD.

## 5. Conclusions

We used an optimized MaxEnt model to predict suitable habitats of *T. minuta* under climate change conditions. Our study concluded that (1) The optimized MaxEnt was highly accurate, and the significant environmental variables influencing the PGDs of *T. minuta* were isothermality, precipitation in the driest quarter, and precipitation in the warmest quarter. (2) Moderate- and high-suitability habitats for *T. minuta* were mainly aggregated in the Beijing–Tianjin–Hebei region, Shanxi, and Yunnan provinces under the current climate. Regardless of the scenario (SSP1-2.6, SSP2-4.5, or SSP5-8.5), the PGDs of *T. minuta* will expand during the 2030s and the 2050s. The conversion of unsuitable habitats to low-suitability habitats and moderate-suitability habitats to high-suitability habitats is significant under climate change. (3) Under the three shared socioeconomic pathways, the centroid of PGDs of *T. minuta* showed a general northward shift in the 2030s and a southward shift in the 2050s. Thus, there is an increasing risk of *Tagetes minuta* L. expanding and invading China, especially in Yunnan, where no occurrence records exist.

## Figures and Tables

**Figure 1 plants-11-03248-f001:**
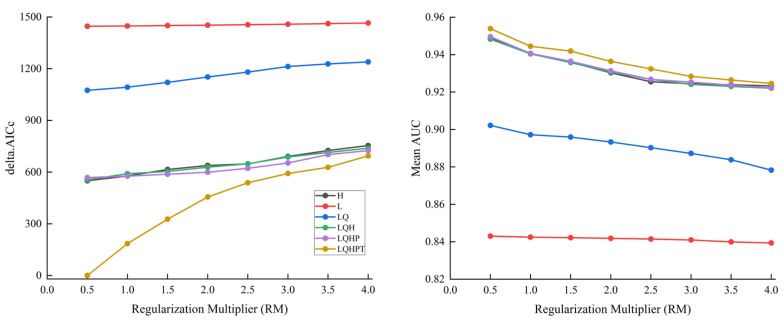
Results of the optimization model under different settings (L: linear; Q: quadratic; P: product; T: threshold; H: hinge).

**Figure 2 plants-11-03248-f002:**
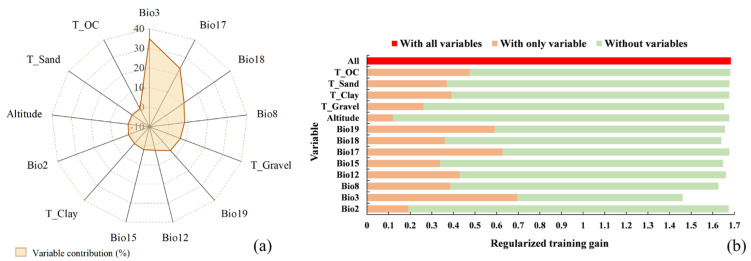
(**a**) Contributions of 13 environmental variables (Bio2: Mean Diurnal Range; Bio3: Isothermality; Bio8: Mean Temperature of Wettest Quarter; Bio12: Annual Precipitation; Bio15: Precipitation Seasonality; Bio17: Precipitation of Driest Quarter; Bio18: Precipitation of Warmest Quarter; Bio19: Precipitation of Coldest Quarter; Altitude; T_Clay: Topsoil Clay Fraction; T_Gravel: Topsoil Gravel Content; T_Sand: Topsoil Sand Fraction; T_OC: Topsoil Organic Carbon); (**b**) Jackknife method results for the environmental variables of *Tagetes minuta*.

**Figure 3 plants-11-03248-f003:**
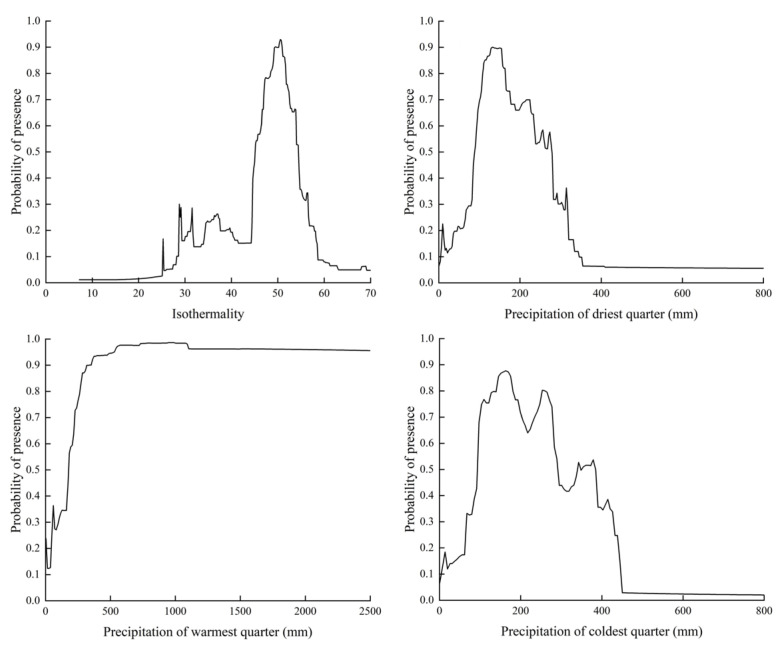
Response curves of presence probability (Bio3, Bio17, Bio18 and Bio19) of *Tagetes minuta*.

**Figure 4 plants-11-03248-f004:**
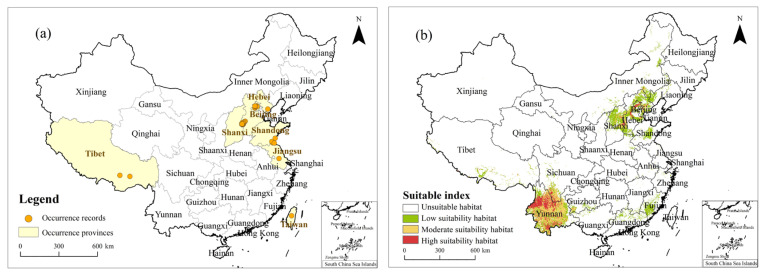
(**a**) Occurrence records of *Tagetes minuta* in China; (**b**) PGDs (Potential Geographical Distributions) of *Tagetes minuta* in China under the current climate.

**Figure 5 plants-11-03248-f005:**
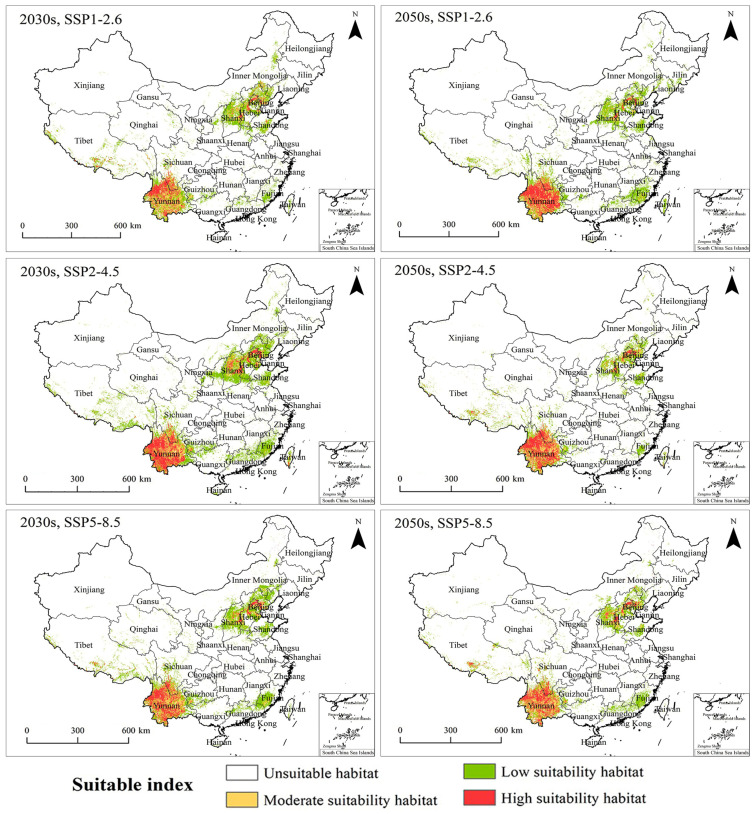
PGDs (Potential Geographical Distributions) of *Tagetes minuta* under different climate change pathways (SSP1-2.6; SSP2-4.5; SSP5-8.5) during the 2030s and 2050s in China.

**Figure 6 plants-11-03248-f006:**
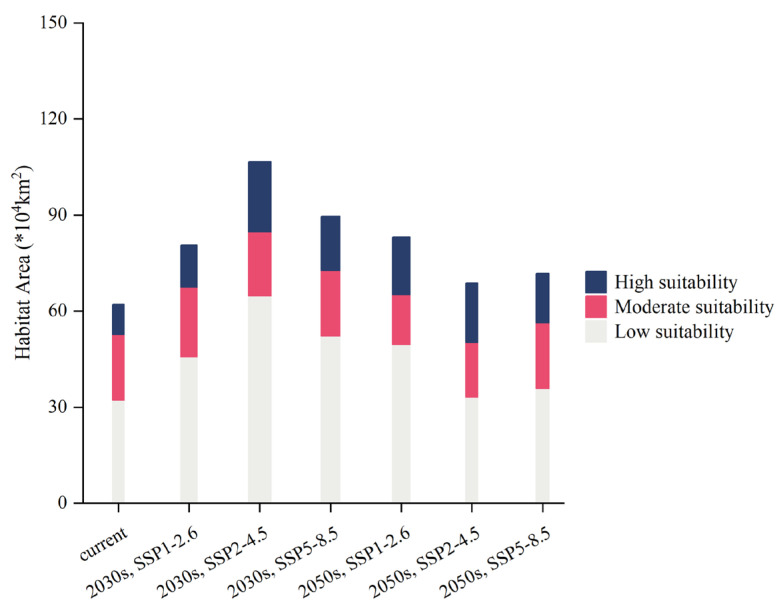
Areas of suitable habitats of *Tagetes minuta* currently and in the future under different pathways (SSP1-2.6; SSP2-4.5; SSP5-8.5).

**Figure 7 plants-11-03248-f007:**
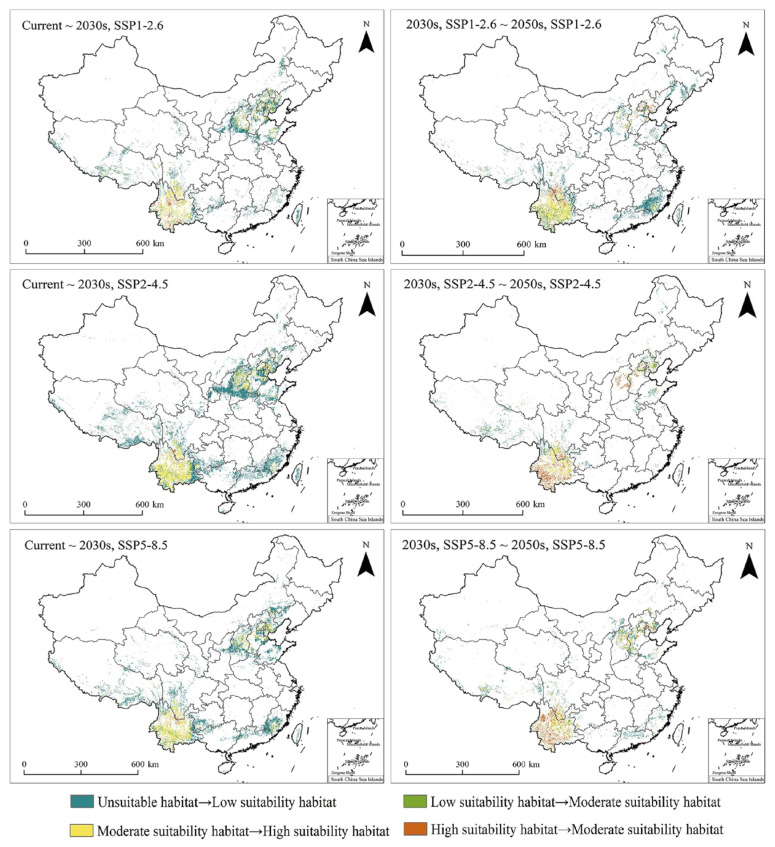
Changes in PGDs (Potential Geographical Distributions) of *Tagetes minuta* under different climate change pathways (SSP1-2.6; SSP2-4.5; SSP5-8.5) from the present to the 2030s and from the 2030s to the 2050s in China.

**Figure 8 plants-11-03248-f008:**
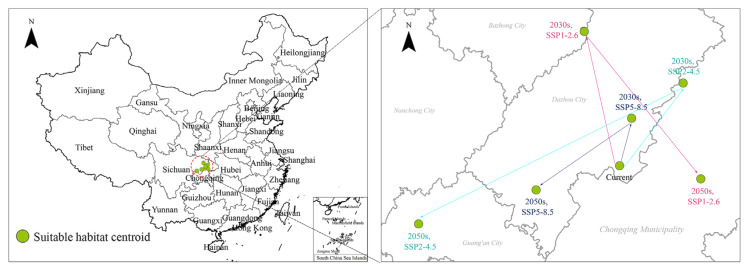
Changes in the centroid distributional shifts of *Tagetes minuta* under climate change.

**Figure 9 plants-11-03248-f009:**
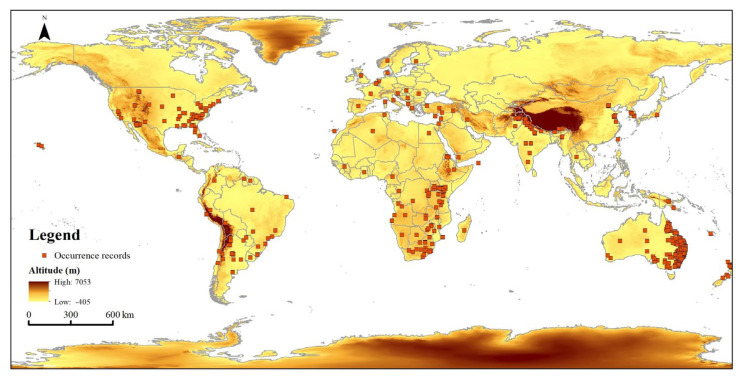
Occurrence records of *Tagetes minuta* L. in the world.

**Figure 10 plants-11-03248-f010:**
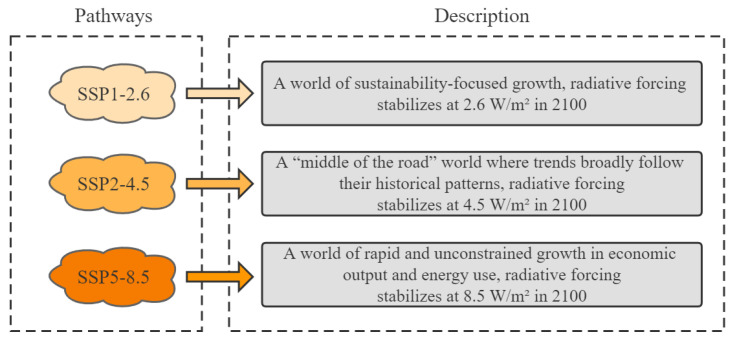
Descriptions of three shared socioeconomic pathways.

## Data Availability

The data presented in this study are available in this article.

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
