# Peer review of "Increased Invasion Risk of Tagetes minuta L. in China under Climate Change: A Study of the Potential Geographical Distributions"

_plants, 2022, doi:10.3390/plants11233248_

Round 1
Reviewer 1 Report
This a very good study almost perfect. The paper is well-written, methodology applied is correct, results and discussion are compatible. I have only some minor issues for authors to consider:
Lines 41-42: there is no linking sentence between description of IAS problem and Asteraceae family. I understand authors’ intention but it sounds artificial. There are some papers showing that Asteraceae have many invasive plants. You can start the sentence “Amongst IAPs Asteraceae…”.
Lines 66-75: the text here is a mixture of introduction and methods. It is not necessary to write which R package was used.
Lines 77-78: Again, it was written what has been done. Here, justification of research should be presented with goals and hypotheses to be tested.
Lines 251-252: “considered a damaging” – if you write that the species is harmful you should provide that is threat to native biodiversity. Write it and give reference for support this statement.
Lines 257-259: This sentence repeats results and is redundant.
Lines 386-387: Write what type of correlation test was used: Pearson, Spearman, Kendall test?
Lines 425-441: This fragment can be shortened. Some sentences are not conclusions but results (line 426-430). I suggest you write it in points
Author Response
Dear Reviewer,
Many thanks for your careful review of this manuscript and for all the valuable comments, which we very much share and to which we have responded on a point-to-point basis. We have now submitted one separate word file of the point-to-point responses and a revised manuscript. If there are any inadequacies in our work, please continue to correct us.
We deeply appreciate your consideration of our manuscript, and we look forward to further processing.
Thank you and best regards.
Yours sincerely,
Yuhan Qi;
Corresponding author:
Name: Wanxue Liu
E-mail: liuwanxue@caas.cn.

Reviewer 2 Report
Journal: Plants
Manuscript ID: plants-1967754
Type of manuscript: Article
Title: Increased Invasion Risk of Tagetes minuta L. in China under Climate Change: A Study of the Potential Geographical Distributions
Submitted to section: Plant Ecology
Comments and Suggestions for Authors:
The paper is based on the principles of data modeling taken from the Global Biodiversity Information Facility to predict the potential geographical distributions of Tagetes minuta L. in China under current and future climate conditions.
The models showed that there is an increasing risk of Tagetes minuta L. expanding and invading China, also in the area (especially in Yunnan), where no occurrence records exist.
ENMTools was used to screen the occurrence records of T. minuta for model simulation. MaxEnt model was optimized using the ENMeval package to obtain the best potential geographical distributions (PGDs) prediction of T. minuta. The calculation algorithm is not well known to me. I assume that the authors are well versed in the given issue.
I didn't understand what the abbreviation AUC is??
What is set to LQPTH? I do not understand the given abbrevation and its meaning
Fig. 1: explaining of Delta AICe is necessary; what does “AUC” mean? I do not understand
Legends of Fig. 4a and 4b are poorly legible
Legends of Fig. 5, 7 and 8 are poorly legible
784 occurrence records and 12 environmental variables were used to predict the potential geographical distributions (PGDs) of T. minuta under current and climatic changes using an optimized MaxEnt model. In addition to the data taken from the databases, did the authors use any of their own observed data?
311-312: B. frondosa are moisture-loving and drought-fearing plants that are more sensitive to precipitation, which is positively correlated with greenhouse gas emissions [52] – I do not understand.
347-348: Support the statement with a citation.
Date of this review: November 16, 2022
Author Response

(The authors gave the same response as above.)
